# Environmental Justice in Industrially Contaminated Sites. A Review of Scientific Evidence in the WHO European Region

**DOI:** 10.3390/ijerph16060998

**Published:** 2019-03-19

**Authors:** Roberto Pasetto, Benedetta Mattioli, Daniela Marsili

**Affiliations:** 1Department of Environment and Health, National Institute of Health, 00161 Rome, Italy; daniela.marsili@iss.it; 2WHO Collaborating Centre for Environmental Health in Contaminated Sites, National Institute of Health, 00161 Rome, Italy; 3National Centre for Global Health, National Institute of Health, 00161 Rome, Italy; benedetta.mattioli@iss.it

**Keywords:** environmental justice, distributive justice, procedural justice, inequalities, inequities, contaminated sites, industrially contaminated sites, industries, socioeconomic status, disadvantaged groups, social capital

## Abstract

In the WHO European Region the topic of contaminated sites is considered a priority among environment and health themes. Communities living in or close to contaminated sites tend to be characterized by a high prevalence of ethnic minorities and by an unfavorable socioeconomic status so rising issues of environmental justice. A structured review was undertaken to describe the contents of original scientific studies analyzing distributive and procedural justice in industrially contaminated sites carried out in the WHO European Region in the period 2010–2017. A systematic search of the literature was performed. In total, 14 articles were identified. Wherever assessments on environmental inequalities were carried out, an overburden of socioeconomic deprivation or vulnerability, with very few exemptions, was observed. The combined effects of environmental and socioeconomic pressures on health were rarely addressed. Results show that the studies on environmental and health inequalities and mechanisms of their generation in areas affected by industrially contaminated sites in the WHO European Region are in their early stages, with exemption of UK. Future efforts should be directed to improve study strategies with national and local assessments in order to provide evidence for equity-oriented interventions to reduce environmental exposure and related health risks caused by industrial contamination.

## 1. Introduction

The Sixth European international WHO Ministerial Conference on Environment and Health, held in Ostrava in 2017, provided a set of suggested actions on seven major themes of environment and health including the one of contaminated sites. The Ostrava declaration promoted a commitment towards “preventing and eliminating the adverse environmental and health effects, costs and inequalities related to waste management and contaminated sites ...” The theme of contaminated sites has been recognized as a priority in Europe from the public health perspective for the first time [1]. This result has been the consequence of growing evidence in high, middle, and low-income countries [2]. In the WHO European Region, (The WHO European Region includes the following countries: Albania, Andorra, Armenia, Austria, Azerbaijan, Belarus, Belgium, Bosnia and Herzegovina, Bulgaria, Croatia, Cyprus, Czech Republic, Denmark, Estonia, Finland, France, Georgia, Germany, Greece, Hungary, Iceland, Ireland, Israel, Italy, Kazakhstan, Kyrgyzstan, Latvia, Lithuania, Luxembourg, Malta, Monaco, Montenegro, Netherlands, Norway, Poland, Portugal, Republic of Moldova, Romania, Russian Federation, San Marino, Serbia, Slovakia, Slovenia, Spain, Sweden, Switzerland, Tajikistan, The former Yugoslav Republic of Macedonia, Turkey, Turkmenistan, Ukraine, United Kingdom and Uzbekistan) wherever assessments have been carried out in the two last decades, a high level of hazardous exposure, and/or an excess of health risk and impact associated to hotspot contaminated areas have been documented [2,3,4].

Recent figures on contaminated sites have been provided by the European Environment Information and Observation Network (EIONET). EIONET includes 27 Member States of the European Union together with Iceland, Liechtenstein, Norway, Switzerland, Turkey, and the West Balkan cooperating countries: Albania, Bosnia, Herzegovina, Croatia, the former Yugoslav Republic of Macedonia, Montenegro, and Serbia, as well as Kosovo under the UN Security Council Resolution 1244/99. EIONET defines ‘contaminated site’ as a well-defined area where the presence of soil contamination has been confirmed and this presents a potential risk to humans, water, ecosystems, or other receptors. The last survey from EIONET, carried out in the years 2011–12, with contribution from countries on a voluntary basis, estimated around 342 thousands of contaminated sites and more than 2.5 million of potential contaminated sites for all the countries belonging to the network [5]. Waste disposal and treatment were estimated to contribute to more than 37% of contaminated sites, while industrial and commercial activities contribute to around 33% [5].

EIONET identified contaminated sites, considering mainly the contamination of soils and the perspective of their remediation. Following a public health perspective, WHO proposed for contaminated sites the following operational definition: “areas hosting or having hosted human activities which have produced or might produce environmental contamination of soil, surface or groundwater, air, and food chain, resulting or being able to result in human health impacts” [6]. Contaminated sites, therefore, can range from areas affected by a single chemical contamination of a single environmental matrix (e.g., the soil contamination caused by a given pesticide) to large areas with soil, water, air, and food chain contamination by multiple chemicals (e.g., the contamination caused by long-term emissions of a petrochemical complex). Under this perspective, the actual or potential risk for human health is the center of research and interventions. The main target populations are communities residing close to contaminated areas, which are “hotspots” of local pollution that can affect all environmental media, especially when still active industries are the sources of contamination.

The European Industrially Contaminated Sites and Health Network (ICSHNet) [7], promoted as an action by the European Cooperation in Science and Technology Association (COST), has focused its activity on contaminated sites with industries as direct or indirect sources of contamination. Activities promoted by the ICSHNet, including representatives of the academia and of public environmental health institutions of 33 countries of the WHO European Region, have provided evidence on environmental and health issues in industrially contaminated sites collecting information from single studies [4] and reviewing available methodologies for health risk, health impact, and epidemiological assessment in those sites [8].

Outside the WHO European Region, the evidence of health risk from industrially contaminated sites has been documented from early nineties, especially in the U.S. among others [9,10,11,12]. In the same years, growing evidence showed that communities living in contaminated sites tended to be characterized by a high prevalence of ethnic minorities and by an unfavorable socioeconomic status [13,14,15]. The topic of Environmental Justice emerged in the U.S. as a theme in the context of contaminated sites in the 1980s as result of grassroots activism of some African America communities fighting against the unfair association between race and poverty and the uneven spatial distribution of waste and industrial sites producing pollution [16]. In the following years, the movement on Environmental Justice broadened its aims and formalized the analysis of inequalities with many new studies that examined the relationship between minority communities, institutional power, and environmental hazards [17]. The theme of Environmental Justice was institutionalized as a central priority of the U.S. Federal Government in 1994 through an Executive Presidential Order. Following this formal endorsement, federal agencies began to include environmental justice considerations in policy implementation and assessment processes [17]. After its birth in the context of contaminated sites, the theme of Environmental Justice broadened its application to a wide variety of environmental themes including their relationship with public health.

The Environmental Justice debate is at its early stages in the European Union (EU) member states and within the European Union institutions [18]. The United Nations Economic Commission for Europe (UNECE) adopted in 1998 the Aarhus Convention on Access to Information, Public Participation in Decision-Making, and Access to Justice in Environmental Matters, which entered into force in 2001 [19]. This legislation framework has contributed in the promotion of Environmental Justice within the WHO European Region [18,20] and has stimulated some remarkable, but sparse, initiatives documenting environmental injustice [21,22].

Regarding Environmental Justice, the U.S. and European contexts are different for several aspects including the major focus in assessing inequities. In Europe, Environmental Justice issues are perceived, analyzed and framed in terms of social categories rather than in racial and ethnic terms. It relies on a different cultural and legal background of public policy in the U.S. and the E.U. [18], but it does not mean that environmental inequalities do not have a racial dimension in Europe as, for example, documented for the Roma community in Central and Eastern Europe [21].

Environmental Justice is commonly recognized having two main dimensions: Distributive Justice and Procedural Justice [23]. Distributive Justice regards the fairness in the distribution of environmental risks and benefits among individuals or population groups (e.g., by ethnicity or socioeconomic status). Procedural justice refers to the mechanisms and processes through which Distributive Justice is created and sustained. These processes rely on institutional and social norms and they include several components such as recognition, capabilities, participation [24,25], and social capital [26], all contributing to decision-making. The lack of these components often affects vulnerable and disadvantaged groups in contaminated areas, characterizing the unfairness of procedural (in)justice for instance by the impossibility to be informed, to express their opinions and their influence in decision-making processes.

Scientific evidence from peer-reviewed studies focused on analyzing the two dimensions of Environmental Justice in the context of contaminated sites in the WHO European Region are apparently scarce. The aim of this contribution is to explore the contents of original scientific studies analyzing the distribution of environmental and health inequalities and mechanisms of their generation in industrially contaminated sites carried out in the WHO European Region in the period 2010–2017.

## 2. Materials and Methods

The review was conducted following a strategy developed by the team of the Department of Social Epidemiology at the IPP, University of Bremen, in the context of systematic reviews on noise, chemicals, air pollution and environmental resources (see respective publications in this special issue). All reviews considered scientific literature published from January 1st, 2010 to December 31st, 2017. The review strategy was developed to carry out reviews on inequalities of the association between socioeconomic and socio-demographic determinants and different environmental health topics in the WHO European Region. The strategy was adapted for this review to deal with the topic of industrially contaminated sites considering inequalities in exposure to such sites or in related health risk (distributive justice), as well as the mechanisms causing and maintaining such inequalities (procedural justice). Evidence obtained with this review will be part of an update of the report “Environmental Health Inequalities in Europe”, which was published by the WHO Regional Office for Europe in 2012 [27]. The topic of contaminated sites was not present in the first report.

The search in literature electronic databases was built up considering three dimensions. The two dimensions of socioeconomic and socio-demographic determinants on one side, and inequalities and inequities on the other side, were considered common among the reviews on environment and health themes, while for the specific dimension of ‘industrially contaminated sites’ a combination of search terms was identified starting from an operational definition of industrially contaminated sites. The following definition adopted by the ICSHNet was taken as reference: “Areas hosting or having hosted industrial human activities which have produced or might produce, directly or indirectly (waste disposals), chemical contamination of soil, surface or ground-water, air, food-chain, resulting or being able to result in human health impacts” [4]. Two categories of search terms were used to explore this dimension: General terms referred to industrially contaminated sites, and specific terms related to the sources of contamination. Three topics were chosen to select the terms related to specific sources of contamination: 1. main heavy industries producing chemical contamination (i.e., metallurgic, chemical, petrochemical, oil refining, steel, gas, and power plants—excluding nuclear plants); 2. mines and quarries; and 3. waste, incinerators, and landfills. These sources were selected considering evidence from the last survey of the European Environment Agency on industrial pollution in Europe [28]. All the identified keywords were combined for research of manuscripts in the three literature databases of MEDLINE (via PubMed), SCOPUS, and Web of Science. The example of the full search strategy applied to PubMed is reported in Table 1.

The eligibility of the studies was assessed on the basis of the abstracts applying the following list of inclusion criteria:Publication in English;Original study (i.e., exclusion of review, editorial, commentaries, studies only with a critical assessment without original data);Study analyzing any kind of socio-demographic or socioeconomic characteristic measured at individual or contextual (area) level (e.g., regional deprivation indices) to assess social inequalities [29];Study analyzing the association between ‘the presence of contamination’/’the assessment of exposure’ and/or health risk/impact due to industrially contaminated sites and socio-demographic/socioeconomic characteristics of individuals or populations residing in areas (i.e., any geographic area, including those with administrative meaning, e.g., regions, provinces, municipalities, districts) meaning the presence/extent of environmental inequalities;A study considering socioeconomic/socio-demographic data not only as confounders in multivariate analysis and indicating quantitative or qualitative results on environmental/health inequalities in the abstract.

The eligibility of the studies was assessed by two reviewers (R.P. and D.M.). Any disagreements between the reviewers over the eligibility of particular studies were resolved by discussion and by consultation of another reviewer (B.M.).

All abstracts of eligible studies were analyzed in order to compare essential characteristics of studies from the WHO European countries with those outside the WHO European Region in the same period. Eligible studies were divided into the two categories of (a) studies documenting inequalities in the distribution of risks related to industrially contaminated sites (i.e., on distributive justice), and (b) studies analyzing mechanisms of their generation and maintenance (i.e., on procedural justice), depending on their main objectives, and study design. Studies on distributive justice were expected to describe the associations between socioeconomic/socio-demographic determinants at individual or group level in areas affected by industrially contaminated sites and/or environmental contamination/exposure from industrially contaminated sites and/or health risks due to differential industrial contamination/exposure. Studies on procedural justice were expected to describe processes and decision-making through which such inequalities are created and sustained.

As a final step, full texts of pertinent studies carried out in the WHO European Region were selected and analyzed in detail by two reviewers, one with expertise in environmental epidemiology (R.P.) of the other in the social sciences (D.M.).

Due to the great heterogeneity of the selected papers concerning study discipline and study design, no standardized quality assessment tool across studies could be applied.

## 3. Results

The articles included in the detailed qualitative analysis were selected through a process described in a Preferred Reporting Items for Systematic Reviews and Meta-Analyses (PRISMA) flow diagram (Figure 1). The literature search identified a total of 453 unique articles. The articles meeting the inclusion criteria were 60. Essential characteristics of these manuscripts, including their classification in studies analyzing distributive or procedural justice, are described in the Appendix A. From the analysis of the abstracts, 45 out of 60 studies appeared to be focused on distributive justice, 14 on procedural justice, and only one seemed to be focused on both dimensions. Fourteen out of 60 studies were carried out in the WHO European Region and therefore were included in the detailed analysis: 10 studies were focused on distributive justice, and four on procedural justice. The other 46 studies were carried out in U.S. (N. 32), Australia (N. 3), China (N. 3), Latin America (N. 3), Canada (N. 2), Ghana (N. 1), India (N. 1), and South Korea (N. 1).

### 3.1. Results on Distributive Justice

Ten studies addressing distributive justice were carried out in the WHO European Region: Two in Scotland [30,31], two in Germany [32,33], one in England [34], one in Czech Republic [35], and four papers in France (in three studies the same areas were analyzed from different perspectives) [36,37,38,39]. The main characteristics and results of selected studies on distributive justice are reported in Table 2.

In the different studies, the unit of analysis taken into account was very heterogeneous and mostly at aggregated level: some were geographical areas of different extension (from hundreds to thousands of residents), some were administrative units (communes, neighborhoods); only in one case, the study was designed using data at the individual level.

The sources of industrial contamination were: industrial plants taken as a general category [30,32,33,39], coal power plants [34,35], incinerators [36,37,38], and landfills [31]. For the industrial plants, the environmental exposure was assessed/operationalized by using European, national, or local registries for air and soil pollution, never collecting original data. For the coal-fields areas, only the location of power plants was considered, whereas for incinerators in France and landfills in the Scottish study, both the location and the amount of emissions were taken into account. Only in the study analyzing landfills, areas affected by high exposure were identified modeling the diffusion of emissions (considering wind speed and frequency of wind) [31].

The socioeconomic or ethnic characteristics considered in most of the studies were education, employment, housing, income, and presence of foreigners. Inequalities in the presence of foreigners were considered in the studies carried out in Germany [32,33] and in French studies on incinerators [36,37,38]. In the German studies, the presence of foreigners was defined as the proportion of inhabitants without a German citizenship, while in the French studies foreigners (recent immigrants) were defined as the foreigners at the time of the census. In France, the concepts of minority and race are unofficial and are not recorded in census data. Therefore, in the French studies, the proportion of persons born abroad was considered among the explanatory variables as a proxy for these dimensions. In some cases, an index of deprivation calculated at area/small-area level considering some socioeconomic domains were used [31,34,39], or constructed ad hoc [30], while in other studies single variables from national census were used [32,33,35,36,37,38]. Only in the study by Riva et al., both socioeconomic and socio-demographic characteristics at individual level and indices of deprivation and social cohesion at area level were used [34].

Results of studies on distributive justice generally show a positive correlation or association between the level of socioeconomic status/deprivation and/or presence of foreigners, and the poor land/air quality or level of contamination. In detail:The studies on industrial soil and air pollution carried out in Glasgow (Scotland) showed a strong positive correlation between growing level of social deprivation and poor land and air quality [31].Of the two studies carried out in Germany, one is focused on the local level [33], while the other provides a national assessment [32]. The first showed that, in the town of Hamburg, toxic release facilities are disproportionately concentrated in, and nearby, neighborhoods with relatively high proportions of foreigners and welfare recipients. The second highlighted a high correlation between percentage of foreigners and exposure to industrial pollution (twice higher in urban areas than rural areas).The Czech study [35] on coalfield areas showed a positive correlation between presence of coal power plants and coal mining and unemployment rate and concentration of ethnic minorities, and a negative correlation with average incomes and pensions and level of education, though the analysis was carried out only at the district level (NUTS 4).The Scottish study on landfills showed that exposure to municipal landfills is concentrated amongst the most deprived areas, and that environmental inequalities around municipal landfill sites have arisen due to a combination of pre-siting and post-siting processes [31].The three French studies regarding incinerators [36,37,38] showed that towns receiving incinerators had higher unemployment and immigrant rates [36] and that a higher proportion of foreigners and persons born abroad in a town is associated with higher odds that the town receives an incinerator. A social gradient was observed with respect to emissions for each of the socioeconomic variables considered: As the proportion of disadvantaged residents increases in a municipality, incinerator emissions also increases [37]. Conversely, in the census period after the opening of the incinerators, no statistically significant effect on employment growth or net migration was observed in populations residing in areas with landfills [38]. The French study on industrial pollution showed that noxious facilities were disproportionately located in higher foreign-born communities. High deprivation also appeared as a predictive factor, although less strongly and less consistently [39].

Only three studies considered and discussed the dimension of health [30,34,35]. Two are mainly focused on environmental inequalities, but report also health data [30,35], while the third one reported only analysis specifically focused on the assessment of environmental health inequalities [34]. This last paper, focused on coalfield areas in England, showed higher odds of reporting less than good health among economically inactive individuals living in coalfield areas in comparison to the same group living in non-coalfield areas. Furthermore, findings showed significant social health inequalities between people living in former coalfield communities that are similar to those observed in non-coalfield areas.

### 3.2. Results on Procedural Justice

The four retrieved papers addressing procedural justice focused on case studies located in France [40] and in Northern European countries: Sweden [41], Finland [42], and Scandinavia/Russia [43]. Details on the characteristics and results of the selected studies are reported in Table 3.

Identified studies made analysis on industrial contaminated areas where the sources of contamination were: mining in two studies [41,43], chemical industry (chemical Seveso plants) [40], and industrial facilities (power plants) [42]. Socioeconomic characteristics considered in the studies were ethnicity, occupation, unemployment, and education.

The papers addressed the following aspects of procedural injustice:Historical misrecognition of the indigenous population as stakeholders living in the contaminated areas, and the lack of influence of indigenous population on decisions concerning land us, in the context of power relationships among involved stakeholders [41,42].Socio-relationships aspects of procedural injustice: communities recognized themselves as poorly informed about the potential impacts and harbored a general distrust of the information provided by the mining companies, corroborated by their inability/impossibility to affect decisions concerning their living environment [43].Roots of environmental injustice related to the decision-making process leading to the choice to locate environmentally burdensome facilities in a disadvantaged area. The long-term siting policies were affected by the poor engagement of inhabitants of that disadvantage district in the decision-making process. In the long period, this modality later led to self-reinforcing siting policies. In other words, environmental injustice was sustained through path-dependent development patterns. The concept of path-dependency was used to explain how environmental injustice was reproduced because of past paths of siting policies locked in subsequent decisions and created a negative twist of accumulating environmental burden [42,43].Analysis of the residents risk perception of the population living in the contaminated areas to explain how cognitive bias within socio-cultural and economic constrains characterized their choice to live in the contaminated area [40].

## 4. Discussion

The results of the systematic review show that the analysis of distributive and procedural justice in industrially contaminated sites available in the peer reviewed scientific literature in the WHO European Region is in its early stages, with the exemption of the UK. Eligible studies were carried out in Northern and Western Europe, with the only exemption of one study in the Czech Republic. The considered sources of industrial contamination were mines (areas with present or former mining activities), industrial plants producing chemical contamination, coal power plants, and incinerators and landfills.

The evidence resulting from the selected studies reflects the geographic-historical pathway of Environmental Justice. When considering identified studies performed outside the WHO European Region, most of them were carried out in the U.S., where the movements on Environmental Justice were born in the 1980s [16]. The amount of articles from the U.S. reflects the cultural background and the sensitivity of the scientific community on Environmental Justice issues also connected with a long lasting grassroots activism of African America communities [16] and wide evidence of racial and ethnic residential segregation [44]. In Europe, the first analysis of inequalities regarding industrially contaminated sites were done in the UK [45] where there is a long tradition in analyzing inequalities in health outcomes by socioeconomic determinants, and where the indices of socioeconomic deprivation at small-area level were developed and used in epidemiological analysis for the first time [46,47]. Therefore, it is not a surprise that among the identified studies, those with a more detailed analysis on inequalities, considering not only environmental inequalities, but also environmental health inequalities, were carried out in Scotland [30] and England [34]. Two of the studies focusing on inequalities were carried out in Germany. They represent the first efforts of assessment of distributive justice at local and at a national level. There is only one study from the Eastern area of the WHO European Region. It is an example of assessment of inequalities associated with the presence of coal industry, though the area level of analysis is too wide (i.e., regional areas—NUTS4 level) for causative inferences [35]. The three studies on incinerators in France were carried out using a similar set of data [36,37,38]. This set of studies allowed to analyze three aspects of distributive justice: If there is inequality, if it is associated to the amount of pollution, and when the inequality was generated (i.e., ante or post location of the polluting source).

Four studies focused on mechanisms (i.e., on procedural justice); they concerned the setting of heavy industries [40,41,42,43]. A common resulting key issue related to procedural (in)justice is the asymmetric power relationships among stakeholders in the decision-making process in which ethnic minorities and/or disadvantaged population sub-groups living in the vicinity of the contaminated areas suffer a lack of influence in decisions concerning the land-use.

### 4.1. Strengths and Limitations

One of the main strengths of this review is that there was no restriction to specific study designs in order to assess scientific evidence on Environmental Justice due to industrially contaminated sites in its broadest way. Furthermore, this review systematically considered a wide range of socioeconomic factors according to the PROGRESS-Plus framework. To the best of our knowledge, this is the first review that systematically assessed scientific evidence on this topic in the WHO European Region. Nevertheless, the picture resulting from the present review has some limitations. First, the identification of studies derived from a combination of key terms related to the three domains of socioeconomic determinants, industrially contaminated sites, and inequalities/inequities. Due to the complexity of the search strategy, it is possible that some potential eligible manuscripts were excluded from the selection because one of the search domains was not present in the abstract.

Second, some potential eligible studies were not included because they were written in a language other than English. In the Identification phase, two original studies carried out in the WHO European Region, one in Italy [48] and the other in Spain [49], with eligible contents on inequalities in the abstract (one of them also including data on health inequalities [48]), were not included in the review because written in their respective national languages. Third, the choice of considering studies carried out in the period 2010–2017, does not allow for an exhaustive picture. In the Screening phase, we found a review published in 2010 exploring inequalities and inequities associated with the location of waste, including evidence from the U.S. and European countries [50]. Finally, the choice of focusing the review on original studies only, have excluded from the selection some studies discussing issues of Environmental Justice in industrially contaminated sites, in particular with contents of procedural justice, without analyzing original data/information.

### 4.2. Critical Analysis of Results

The selected studies on inequalities were based on geographical analysis reporting details on the association between the presence of industrial sources of contamination and the disproportion of socioeconomic vulnerabilities in the most affected areas. Only one manuscript is published in a journal specifically devoted to Environmental Justice issues [37]. Three studies report figures on health [30,34,35], with one of them reporting analysis directly allowing environmental health inequality inferences [30,34]. Assessing inequalities and inequities on industrially contaminated sites means dealing with local communities (i.e., communities located in the neighborhood of polluting sources and exposed to the contaminants). The assessments can be directed to evaluate a phenomenon associated to the spatio-temporal distribution and quantification of contaminants. In this case, the main challenge is in identifying areas, and related populations, at differential exposure. In other cases, the main focus is in identifying populations affected by relevant phenomena in terms of area units with an administrative identity (e.g., communes in France [35], municipalities in Italy [48]). In this latter case, the focus is on identifying meaningful local administrative units, and their relative populations, with differential influences (including the exposure to contaminants) from a given source of contamination. The choice of the unit of analysis makes possible to identify local authorities that can be eventually involved in the interventions to reduce inequalities.

Methods used in the selected studies for the qualitative and quantitative assessment of inequalities were very heterogeneous reflecting the differences in study design and data availability. They range from bivariate analysis to assess the correlation between the presence/absence of industrial sites and the socioeconomic level to multivariate analysis using different regression models for the assessment of associations between exposure and multiple socioeconomic/socio-demographic determinants. The models applied include econometric spatial models to incorporate the spatial dependence in the analysis [32] as well as multilevel models used in a study at individual level with socioeconomic characteristic attributed using both individual and area level data [34]. In one study the method of differences in differences was applied to verify if inequalities were present before and/or after the sitting of polluting sources [38].

Methodological issues in identifying the appropriate scale (i.e., area level) of analysis, in verifying their differential exposure, and in analyzing the associations between exposure and socioeconomic status have been described among the others by Mitchell and Walker, Chakraborty, and Mennis and Hackert [20,51,52].

In the assessment of environmental inequalities in industrially contaminated sites, studies carried out at different scales in a country can give complementary information. On one side, local studies with collection of *ad hoc* information on exposure and socioeconomic determinants can provide both evidences on the distribution of inequalities and on their mechanisms. On the other side, studies with a national basis, can contribute in understanding common phenomena at the national level and verifying if these phenomena have different spatio-temporal patterns within country. In the WHO European countries, this complementarity focusing on the same source of contamination (e.g., industrial facilities) seems to have been explored in the UK and very recently in Germany, while it is commonly assessed by different studies in the U.S.

Considering a public health perspective, major efforts should be directed at including the analysis of the health dimension. In the Screening phase of the present review, some studies placed in the WHO European Region were excluded from the analysis because socioeconomic factors were included only as confounders in the assessment of associations between industrial pollution and health outcomes: This is the case of two studies carried out in Poland [53] and Belgium [54]. The availability of these data means that those countries have data and potentiality to explore environmental health inequalities in industrially contaminated areas.

The socioeconomic/socio-demographic attributes of populations affected by industrial pollution in the retrieved studies were assessed using single variables (usually from national census or from data of local bureau of statistics) or combining variables in indices of multiple deprivation. Only in one study an ad hoc index of social capital was built up [34]. More efforts should be directed in combining information retrieved and attributable to individuals with those representing the context, especially in environmental health inequality studies. In fact, individual and contextual socioeconomic determinants can have different influences on health outcomes as it was observed in epidemiological multilevel studies carried out in some European countries [55,56].

More efforts should be directed to assess not only inequalities related to socioeconomic determinants, but also those associated to ethnicity. This aspect requires both intervention at regulatory level and at a research level. It is necessary to improve the kind of available information or to collect *ad hoc* data able to address such dimension. The use of information on foreigners or persons born abroad as proxies for minority and race is not sufficient. Such proxies are weak and completely miss second-generation immigrants (and the proportion of ethnic population they represent).

Studies focusing on the mechanisms of generating procedural injustice were centered on historical and socio-environmental analysis relying on a qualitative research approach. This appears particularly effective to increase the understanding of the roots and reasons of environmental procedural injustice occurred and still occurring in each industrially contaminated site. The tools used in these studies were semi-structured interviews and focus groups as well as the study of historical documents [40,42,43]. They provided information on the impossibility for the inhabitants to be recognized as stakeholders participating in the decision-making processes concerning land use and environmental and socio-cultural degradation of their territories. The misrecognition of the ethnic minorities or disadvantaged sub-groups of population living in the vicinity of industrially contaminated sites is often historically maintained from the construction of the industrial setting to its operation during a relatively long time window [41]. In fact, in terms of procedural justice, sub-groups of people have been historically excluded or marginalized by the institutions—at all scales, from the local to the global—which are responsible for developing policies and taking decisions changing environmental conditions of the areas where they live [57].

The study by Flanquart et al. also relied on the individual perception of inhabitants concerning the severity of the contamination of their living environment and their powerlessness to leave the polluted territory for building a new healthy life away from there [40].

It is important to highlight that the socioeconomic variables commonly used to assess the socioeconomic status of the residents in industrially contaminated sites are unable to account for the social dimensions, such as the quality of relationships among the involved stakeholders, the existence of local communication networks as well as of participative processes in the decision-making. In the study by Suopajärvi e coll. [43] social impacts affecting the people living in the mining area were described according to the definition provided by the International Association for Impact Assessment [58]. Under this definition, the social impacts are intended as those experienced in various spheres of life, such as culture, community, political system, environment, health, way of life, personal/property rights, fears, and aspirations.

### 4.3. Studies Outside the WHO European Region

Essential characteristics and results of studies carried out outside the WHO European Region in the same period of those identified for the WHO European Region are reported for comparison.

The analysis of abstracts of studies focusing on distributive justice worldwide showed that U.S. is the country with the largest number of published studies (N. 24, including two studies in the U.S.-Mexico borderlands), as expected [59,60,61,62,63,64,65,66,67,68,69,70,71,72,73,74,75,76,77,78,79,80,81,82]. Among the studies published outside the WHO European Region and not in the U.S., two were carried out in Australia [83,84], three in China [85,86,87], two in Latin America [88,89], and one in Canada [90], Ghana [91], India [92], and South Korea [93]. Studies carried out in the U.S. not only are more numerous, but represent a wide variety of study design: (i) the analysis were both at the national level and at local scale, in this latter case allowing inferences at different area/population levels (county, city/town, census block, community); (ii) the spatial distribution of sources of contamination was identified usually together with a quantification of emissions and in some studies exposure was modeled; and (iii) in some studies the dimension of health was considered and in one study both individual and community data, allowing at the same time the assessment of both inequalities and their mechanisms, were used [53]. In most of the studies from the U.S., inequalities were assessed for ethnicity, which seems to be their most predictive factor. Studies placed in other countries, with the exemption of Australia and Canada, are simpler in design, which is mainly due to limits in the spatial scale data availability.

The analysis of abstracts of studies focusing on procedural justice worldwide also reveals that the U.S. is the country with the largest number of published studies, thus corroborating the historical pathway of Environmental Justice in that country. Among the 15 studies published outside the WHO European Region, eight concern procedural justice in contaminated sites in the U.S. (including one in the U.S.-Mexico border) [60,94,95,96,97,98,99,100,101], one in Australia [102] and one in Latin America [103]. Comparing results of the selected studies within and outside the WHO European Region, the inequity issue has the same main cause: The mis-recognition of the identity and rights of ethnic minorities and disadvantaged communities (socioeconomic vulnerability) by the Governmental Authorities and/or the foreign private companies in the decision-making processes with the use of the land where they live. This increases the impact on the communities that are disproportionately affected by both the environmental contamination and degradation of socio-cultural dimensions of their territory.

## 5. Conclusions

The original studies published in the WHO European Region in the period 2010–2017 are few, but express a growing awareness on the theme of Environmental Justice in the context of industrially contaminated sites. Wherever assessment on environmental inequalities were carried out, an overburden of socioeconomic deprivation and vulnerabilities, with very few exemptions, was observed. The combined effects of environmental and socioeconomic pressures on health were rarely addressed. Due to the limits in the search strategy, this review should be considered as exploratory on the available evidence at the WHO European level and it should be integrated with a detailed analysis in each country.

The evaluation of the retrieved studies on procedural and distributive justice, highlights at least three directions for future studies. The first is to develop study strategies that include different phases and study methods, with the contribution of experts of social, environmental, and health sciences, in order to improve the causative assessment of environmental health inequalities. Such assessment is the basis to plan interventions able to result in long-term solutions [104]. Furthermore, any judgment of inequity (i.e., on distributive injustice) should be sustained by an assessment of the causal nature of inequalities. The second is to improve applications and study designs in order to assess not only environmental inequalities, but also to include the dimension of health in the analysis. The third is to consider both local assessments and national assessments. Local assessments can provide evidence and information with more details useful for local interventions, while national assessments can give general information useful to identify priorities (for example by identifying regions with more inequalities and unfairness) for the management of inequalities and inequities at the national level.

The contribution of single studies can be integrated with evidence of *ad hoc* national epidemiological monitoring programs of communities affected by industrial contamination [105]. Resulting overall evidence together with tailored communication plans [106], envisaging stakeholder engagement, can contribute to further enhance the promotion of public health from an Environmental Justice perspective in industrially contaminated areas.

## Figures and Tables

**Figure 1 ijerph-16-00998-f001:**
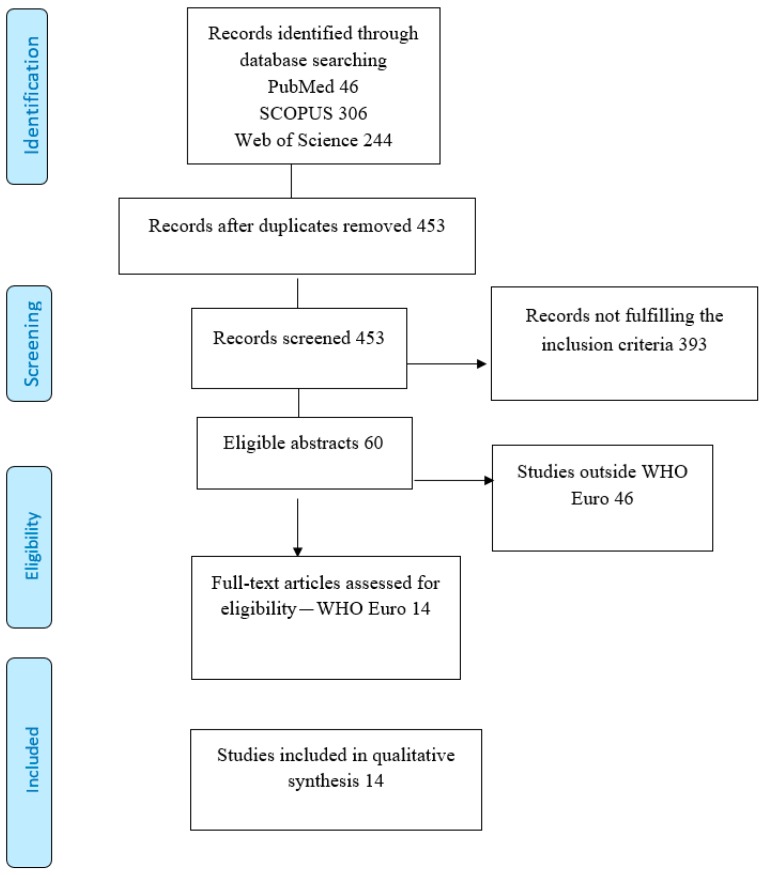
Preferred Reporting Items for Systematic Reviews and Meta-Analyses (PRISMA) flowchart of the literature search.

**Table 1 ijerph-16-00998-t001:** Full search strategy executed on PubMed/Medline database.

Search Domain	Query
#1—Socioeconomic and sociodemographic determinants	(sociological factors[MeSH Terms] OR disadvantaged[All Fields] OR disadvantage[All Fields] OR deprived[All Fields] OR social[All Fields] OR socio * [All Fields] OR vulnerable populations[MeSH Terms] OR vulnerable[All Fields] OR vulnerability[ALL Fields] OR psychosocial[All Fields] OR psycho-social[All Fields] OR socioeconomic factors[MeSH Terms] OR socio-economic[ALL Fields] OR deprivation[All Fields] OR socio-demographic[All Fields])
#2—Industrially contaminated sites	(industrial pollution prevention and control sites[Title/Abstract] OR IPPC[Title/Abstract] OR european pollutant emission register[Title/Abstract] OR contaminated land[Title/Abstract] OR contaminated site * [Title/Abstract] OR industrial site * [Title/Abstract] OR industrial pollution[Title/Abstract] OR industrial water pollution[Title/Abstract] OR industrial air pollution[Title/Abstract] OR industrial soil pollution[Title/Abstract] OR superfund[Title/Abstract] OR industrial facilities [Title/Abstract] OR ((industry * [Title/Abstract] OR site[Title/Abstract] OR plant * [Title/Abstract]) AND (steel [Title/Abstract] OR iron [Title/Abstract] OR metallurgic * [Title/Abstract] OR chemical [Title/Abstract] OR petroleum * [Title/Abstract] OR petrochemical * [Title/Abstract] OR oil refinery[Title/Abstract] OR steel[Title/Abstract] OR gas[Title/Abstract] OR power plant[Title/Abstract] OR mining[Title/Abstract] OR quarr * [Title/Abstract] OR waste[Title/Abstract] OR incinerator * [Title/Abstract] OR landfill * [Title/Abstract]))
#3—Inequalities and inequities	(inequality[Title/Abstract] OR inequity[Title/Abstract] OR inequities[Title/Abstract] OR inequalities[Title/Abstract] OR unequal[Title/Abstract] OR environmental justice[Title/Abstract] OR environmental injustice[Title/Abstract])
#4—Period	(“2010/01/01”[Date—Publication]: “2017/12/31”[Date—Publication])
**Final search**	#1 AND #2 AND #3 AND #4

* the use of the asterisks in PubMed allow to consider each declination of the word to which they are associated.

**Table 2 ijerph-16-00998-t002:** Main characteristics and results of identified studies on distributive justice in industrially contaminated sites, 2010–2017.

Ref.	Type of Contamination	Country	National/Local	Study Design and *Analysis*	Unit of Analysis	Exposure Assessment	Socioeconomic Characteristics/Social Dimensions
[30]	Soil metal content, air pollution	Scotland.	Local (Glasgow)	Small-area study *Bivariate analysis: Person’s correlation coefficient*	Aggregated level: areas including 4000 households (Intermediate Geography Zone)	Level of heavy metals in soil and concentration of NO_2_ and PM10 in air. Index of pollution at area level	Index of multiple deprivation composed by six domains: education, employment, housing, income, access to services, crime
**Results on environmental inequalities:** Strong positive correlation between growing level of deprivation and poor land and air quality.							
**Results on health inequalities:** Positive correlation between Standardized Incidence Ratio (SIR) for respiratory diseases and soil and air pollution and growing level of deprivation. Significant negative association between least deprived categories and SIR for respiratory diseases.							
[34]	coalfield areas	England	National	Cross-sectional Data on health outcomes and confounders at individual level, data on socioeconomic variables both at individual and area level (contextual) *Multivariate analysis: multilevel logistic models*	Individual level: annual representative cross-sectional survey of the English population	Living in a former coalfield area	Individual level: marital status, economic activity, occupation and social class Contextual level: Index of Multiple deprivation and index of social cohesion
**Results on environmental inequalities:** All analysis includes the assessment on health (see column ‘results on health inequalities’).							
**Results on health inequalities:** Higher odds of reporting less than good health among economically inactive individuals living in coalfield areas in comparison to the same group living in non-coalfield areas. Significant social health inequalities between people living in former coalfield communities are similar to those observed in non-coalfield areas.							
[35]	coalfield areas	Czech Republic	National	Ecological study *Bivariate analysis: Person’s correlation coefficient*	Aggregated level: districts (NUTS4)	Presence of coal power plants	Socioeconomic variables associated with the domains of life quality, labor market, social capital, and social cohesion
**Results on environmental inequalities:** Positive correlation between the presence of coal power plants and coal mining and unemployment rate and concentration of ethnic minorities. Negative correlation with average incomes and pensions and level of education.							
**Results on health inequalities:** Association between the presence of coal power plants and coal mining and higher rates of abortion, higher infant mortality, and lower male life expectancy.							
[33]	Industrial pollution	Germany	Local (Hanburg)	Small-area study *Bivariate: Person’s correlation coefficient* *Multivariate analysis: Ordinary Least Squared model*	Aggregated level: neighborhood	Location of industrial facilities	Proportion of foreigners and population receiving public assistance
**Results on environmental inequalities:** Toxic release facilities are disproportionately concentrated in, and nearby, neighborhoods with relatively high proportions of foreigners and population receiving public assistance.							
[32]	Industrial pollution	Germany	National	Small-area study *Multivariate analysis: Ordinary Least Squared model and Spatial Model (SLX)*	Aggregated level: areas containing an average of 778 inhabitants	Location of industrial facilities and categorization of their emissions	Proportion of foreigners and vacant houses; living spaces
**Results on environmental inequalities:** High correlation between percentage of foreigners and exposure to industrial pollution. Population density of the surrounding area is a significant predictor of pollution only in urban areas. The percentage of vacant houses correlates with pollution only in rural areas							
[39]	Industrial pollution	France	Local (Franche-Comte’ region)	Small-area study *Multivariate analysis: Bayesian hierarchical logistic regression*	Aggregated level: areas with a mean population of 569 (IRISes)	Location of industrial facilities (areas whose borders intersected circles with a radius of 2 km from industrial facilities)	Index of multiple deprivation composed by four domains: unemployment, house ownership, car ownership, overcrowding Persons born abroad
**Results on environmental inequalities:** Noxious facilities are disproportionately located in higher foreign-born communities after controlling for deprivation, population density and rural/urban status. High deprivation also appears as apredictive factor, although less strongly and less consistently.							
[36]	Incinerators	France	National	Ecological study *Descriptive analysis* *Multivariate analysis: Spatial logistic regression*	Aggregated level: communes	Presence or absence of an incinerator in the communes	Unemployment rate, proportion of foreigners and person born abroad
**Results on environmental inequalities:** Towns receiving incinerators had higher unemployment and immigrant rates.							
[37]	Incinerators	France	National	Ecological study *Multivariate analysis: Multilevel linear models with random effects*	Aggregated level: communes	Total annual emissions from incinerators in communes with more than one incinerator	Proportion of unemployed people, immigrants, and persons born abroad
**Results on environmental inequalities:** A social gradient was observed with respect to emissions for each of the three considered socioeconomic variables.							
[38]	Incinerators	France	National	Ecological study *Differences in differences*	Aggregated level: communes	Presence or absence of an incinerator	Unemployment rate proportion of foreigners and person born abroad
**Results on environmental inequalities:** Incinerators had no statistically significant effect on employment growth or net migration of the established population in the census period after they opened.							
[31]	landfills	Scotland	National	Small-area study *Multivariate analysis: Ordinary least squares regression and Logistic regression*	Aggregated level: areas with approximately 500 persons (Continuous Areas Through Time)	Air pollution from landfills in each area modeling exposure using a landfill exposure index incorporating site specific emissions and local wind conditions	Index of multiple deprivation composed by: lack of car ownership, low occupational social class, overcrowded household, and male unemployment
**Results on environmental inequalities:** Exposure to municipal landfill in Scotland is concentrated amongst the most deprived areas. Environmental inequalities around municipal landfill sites in Scotland have arisen due to a combination of presiting and postsiting processes.							

**Table 3 ijerph-16-00998-t003:** Main characteristics and results of identified studies on distributive justice in industrially contaminated sites, 2010–2017.

Ref.	Type of Contamination	Country	National/Local	Study Design and *Methods*	Unit of Analysis	Socioeconomic Characteristics/Social Dimensions
[40]	Heavy industry: Chemical (Seveso plants)	France	Local (Mardyck village within the urban area of Dunkirk)	Socio-environmental study *Interviews*	Individual level: Adults (People aged at least 16); N. 158 as fraction of the total population N. 270	Occupation (Socio-professional class and Unemployment), Education, house ownership
**Results on environmental inequalities:** Economic and social contains, cognitive and cultural bias. Quite low educational level and intermediate professional levels. Higher unemployment than in France average. Perception of residents: the majority emphasized the availability of quiet and pleasant public space, while a minority declared lack of choice relating to living in the village due to economic constraints (unable to move from the village).						
[41]	Mining	Sweden	Local (Gállok, area in Jokkmok municipality)	Socio-environmental study *Interviews, unstructured non-participant observations and documents*	Individual level different stakeholders; N. 13	Ethnicity (Sami indigenous population)
**Results on environmental inequalities:** Asymmetric power relations among stakeholders. Historical mis-recognition of the indigenous population (Sami) as relevant stakeholder, resulting in lack of influence in decisions concerning land-use.						
[42]	Industrial facilities (power plants, waste disposal)	Finland	Local (Helsinki, Sörnäinen district)	Historical analysis *Information from archival sources and documents*	-	Socioeconomic class
**Results on environmental inequalities:** Social and environmental living conditions were poor. Disadvantaged area. Environmental inequities were due to land-use. Siting decisions and related decision-making processes resulted in a trend of accumulating environmental burden (self-reinforcing siting policies).						
[43]	Mining	Northern Europe	Local (8 communities living in areas around contamination source in Norway, Sweden, Finland and Murmansk region in Norwest Russia)	Socio-environmental study *Interviews and focus groups*	Individual level. Different stakeholders; N. 85	Socioeconomic status associated with cultural values (i.e., way of life)
**Results on environmental inequalities:** Inhabitants of the communities have no power to influence the development of the areas where they live in respect to State-led or international companies (glocal phenomena). Lack of information and participation in decisions. Loss of their own cultural way of life.						

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
