# Peer review of "Environmental Justice in Industrially Contaminated Sites. A Review of Scientific Evidence in the WHO European Region"

_ijerph, 2019, doi:10.3390/ijerph16060998_

Round 1

Reviewer 1 Report

This is a timely, thorough, and well written study and report. The need is established, literature reviewed, results and findings are complete and meaningful, and recommended directions for further study helpful.

Author Response

We wish to thank the reviewer for such gratifying and encouraging review

Reviewer 2 Report

This article has significant framing issues in that it relies in the history/framing of environmental justice in the United States without sufficient understanding (or citations) (lines 79-84) of that long history that covers both social movements and the associated scientific/public health research that has emerged on EJ for the past four decades. They cite only one historical article by Bullard while they note the work has progressed, they do not demonstrate a grasp on this body of research.  Yet, they use EJ as a framing for the article itself-- that is situated in the EU and WHO EU. The US movement  is first and foremost a movement for racial justice. The effort to map this onto the EU context without any explanations of the difference in social movements and related regulatory science makes it unclear as to the purpose of this literature review. The authors note that the analysis of procedural and distributive justice is limited in the EU context and could look at what is gathered and why as well as the data limitations that make a racial justice analysis difficult. Thus, it is overall unclear what this literature review hopes to contribute to scientific, public health, or policy realms.

In addition, the authors do not note that in countries such as France, no demographic data on race is allowed to be collected. If US environmental justice research has taught us anything, it is that income and "foreign born" are not a proxy for race. Authors note limits on data availability but do not consider the extent of these limits, particularly as they argue for an approach that is EJ relevant. This is perhaps untenable in the EU context given data availability. Lines 102-104 note the scarcity of research but it would perhaps be useful to think about parallel or different social movements in EU and note the links back to the state and or/regulation.  If there are no clear EJ policies in EU, what is the relevance of this analysis? The research sorts on socioeconomic status and "foreign born" -- might think through the relevance of these categories and why/how they are used-- and what sorts of framings might be more productive.  

The chart on lines 183-214 is sparse.  It should be able to integrate both the number of articles and the process as well as the criteria.  It should be able to stand alone.  As it is currently, it does not provide much useful information and could both take up less space or be removed altogether.  It would be useful if criteria data is included in the method for narrowing.  

Lines 232-238: The authors note the socioeconomic and ethnic characteristics for inclusion in studies but do not discuss why these are typically included versus other metrics in the discussion.  The authors need to overall better link theoretical framings to the empirical data.  It is relevant that there has not been attention to race in the EU in terms of procedural and distributive justice vis a vis hazards-- and that is perhaps a more worthwhile contribution than the current one that can borrow from the EJ analysis in the US but is more rooted in local realities. 

The Tables are difficult to read.  Might be reformatted to landscape. 

Lines 333-337 note asymmetric power relationships.  This might be a more useful framing. 

The work by Barbara Allen is not referenced from her work in France. Her research might be a useful citation for your work.  

Given the diversity of languages in the EU, the lack of integration of non-English publications seems like a potential major limitation to this project.  Further, it is unclear why the EU is chosen as a site of analysis rather than single countries.  Is it that the regulatory apparatus spans the EU and this is more critical than within country efforts in terms of hazardous waste and undesirable land uses?  This needs to be both explained and justified. 

The authors assert on lines 457-459 that inequity has a single main cause but they do not provide any citations to back this up. I urge them to consider looking into the social science literature if they could not find anything in the scientific literature. 

Overall, this type of literature review could be a contribution to the field if it was framed in a way that was both clearer and more relevant to ongoing issues of distributional injustice in Europe or within single countries in Europe.  Given a major revision and reframing with relevance to community led efforts in EU, the current regulatory structures in EU, and a clear sense of the realities of collecting racial data-- it would be a more valuable contribution. 

Author Response

We have responded to reviewer's comments in the uploaded Word file

Reviewer 3 Report

This article is a valuable contribution to the EJ literature. My comments are below.

Line 80 – put space in between 80s as

Line 152 and 167 – are these two reviewers also the authors of this paper? Please clarify who they are.

Lines 137-151 - Rephrase the criteria so they are criteria for inclusion, not exclusion. For example, publication in English, original study, study analyzing any kind of…  This will be in agreement with Table 1

Footnote 1 – move it earlier in the text; to line 71, when the WHO European Region is mentioned for the first time

Line 174 – what is PRISMA? Please explain

Line 181 – add here what regions were the other 43 studies about

Line 233 – give definition of “foreigners”. For example, are these foreign-born residents or (il)legal migrants or refugees?

Table 2 – add more details about methods of analysis. Just saying “bivariate” or “multivariate analysis” is not enough.

Line 278 – France is not considered a Northern European country

Line 298 – “created”

Section 4.2 – this section needs a detailed  discussion of analytical methods used in these papers. Again, just saying “bivariate” or “multivariate analysis” is not enough. Give more details – for example, were these OLS or logistic regressions, or generalized linear models etc.

Line 361-364 – rephrase these two sentences so these parts are fixed;  “…had  in  the  geographical  analysis  their  main  focus” and “  … Journals  have  in  the  geographical  analysis  their  main  aim”

Line 419 and other lines – change “e coll” to “et al.”

Lies 448-450 – rephrase this sentence. It is not clear what you mean by “limits in the spatial scale data availability.”

Author Response

We have responded to the reviewer's comments in the uploaded file

Round 2

Reviewer 2 Report

The revision is sufficient. The use of the term "environmental justice" is of continued concern, given the very different structural realities of the literature you cite and the general assertions made about the EU.  "Environmental inequality" is likely a more reflective term.  It would be useful to contextualize this for future research.